**Investigation**

# The role of toxin/antidote genes in the maintenance and evolution of accessory chromosomes in *Fusarium*

Linnea Sandell,[1,*] Adrian Forsythe,[1] Anna Mirandola,[1,2] Samuel Jorayev,[1] Andrew S. Urquhart,[1] Alexandra Granger Farbos,[3] Sven J. Saupe,[3] Aaron A. Vogan [1,*]

[1]Department of Organismal Biology, Uppsala University, Uppsala 752 36, Sweden
[2]Department of Biotechnology, University of Verona, Verona 37134, Italy
[3]Institut de Biochimie et de Génétique Cellulaire, CNRS, University of Bordeaux, Bordeaux Cedex 33077, France

*Corresponding authors: Linnea Sandell, Department of Organismal Biology, Uppsala University, Uppsala 752 36, Sweden. Email: nhl.sandell@gmail.com; Aaron A. Vogan, Department of Organismal Biology, Uppsala University, Uppsala 752 36, Sweden. Email: aaron.vogan@ebc.uu.se.

The genomic diversity of many fungal species is augmented by accessory chromosomes, which are variably present in individual strains. These genomic regions evolve rapidly, accumulating genes important in pathogenicity but also harbor a significant number of transposable elements. This duality suggests a trade-off: accessory chromosomes provide infection-related benefits while otherwise being deleterious due to their highly repetitive nature and contributions to genomic instability. Despite this, accessory chromosomes often appear to be stably maintained even when strains are grown on media, with no plant host. Previously, we had observed that genes homologous to meiotic drive toxin/antidote proteins from *Podospora anserina* (*Spoks*) are abundant on accessory chromosomes in various *Fusarium* species. Using a functionality screen in yeast, we demonstrate that some of these homologs have active toxin and antidote properties. We propose that these selfish genes could maintain accessory chromosomes during vegetative growth and may influence their spread via parasexual cycles. Finally, as *Spok* genes are found on the newly described transposable element superfamily *Starships*, we also present a model for how these transposable elements could play a role in forming accessory chromosomes and regions. These results illuminate a mysterious facet of fungal biology, a key step towards describing the origin, spread, and maintenance of pathogenicity in many fungal species.

Keywords: Fusarium; toxin–antidote; gene drive; Podospora

## Introduction

Fungi are among the most devastating plant pathogens and represent one of the largest threats to food security (Steinberg and Gurr 2020). Efforts to combat and control fungi are constantly fraught with challenges from novel strains and/or host jumping (Fones et al. 2020). Part of the reason that fungi are such difficult pests to manage is their genomic diversity. Of particular notoriety are those strains which possess accessory chromosomes (ACs). In addition to a set of "core" chromosomes, shared by all members of a given species, many isolates carry a number of additional chromosomes. These extra chromosomes are referred to as lineage specific, conditionally dispensable, or accessory (among others), and were first described by their resemblance to B chromosomes found in animals and plants (Miao et al. 1991). These ACs have high evolutionary rates, low gene density, differential codon usage, and large repeat content (Coleman et al. 2009; Ma et al. 2010; Goodwin et al. 2011; Yang et al. 2020). In addition to a large number of transposable elements (TEs), they also accumulate genes beneficial to the fungi. This has been demonstrated in pathogenic fungi, where effector genes that help the fungi infect its host are often found in accessory parts of the genomes (van Dam et al. 2016; Henry et al. 2021). Due to the contrasting burden of TEs and benefits of virulence-related genes, ACs are

assumed to constitute a trade-off between genomic stability and pathogenicity.

Contrary to the hypothesis that ACs are maintained because of benefits during fungal colonization/infection of plants, some ACs seem to be maintained without contributing advantages to the host, persisting even when grown on permissive media where the benefits of pathogen-specific genes have little or no relevance for growth. For example, in *Fusarium oxysporum* f. sp. *lycopersici*, the ACs are lost only in every 35 000th asexual spore (Vlaardingerbroek et al. 2016). Thus, while ACs in this species are indeed tightly linked to virulence (Ma et al. 2010), there appears to be little selection to lose ACs when this selective pressure is removed. In contrast, in the wheat pathogen *Zymoseptoria tritici* the loss of ACs occurs more frequently, approximately once in every 50th asexual generation (Möller et al. 2018). However, the connection between virulence and ACs in *Z. tritici* is more complex than in *F. oxysporum*, with some ACs apparently inhibiting virulence on certain cultivars (Habig et al. 2017). Neither of these patterns strictly support a role of trade-offs in maintaining ACs in pathogenic fungi. As such, other mechanisms must be investigated.

Here, we hypothesize that the maintenance and spread of ACs in species of the genus *Fusarium* are linked to the presence of homologs of selfish spore-killing elements that were first reported in *Podospora*, known as *Spok* genes (van der Gaag et al. 2000;

Grognet et al. 2014). This hypothesis arises from compelling observations from previous work. First, large numbers of *Spok* homologs are found on ACs in *Fusarium* (Vogan et al. 2019). This includes chromosomes that exhibit whole chromosome transfer between strains of *F. oxysporum* and that have strong contributions to virulence (Ma et al. 2010; Shahi et al. 2016). Second, in addition to their role in the meiotic cycle of *P. anserina*, the *Spok* genes are expressed during vegetative growth and exhibit strong toxicity (Vogan et al. 2019). Furthermore, the *Spok* genes are capable of inducing cell death across a vast breadth of phylogenetic diversity, including other sordariales (Grognet et al. 2014), the model yeast *Saccharomyces cerevisiae*, and even the bacterium *Escherichia coli* (Urquhart and Gardiner 2023). This suggests that their toxic function is not limited to the sexual cycle, nor by genome or species-specific context. Therefore, the SPOK proteins could act during vegetative growth in *Fusarium* to prevent the loss of accessory chromosomes that carry them.

The precise mechanism of action of the *Spok* genes has yet to be uncovered, but a number of important characteristics have been discerned. Four *Spok* homologs are known from *Podospora*, *Spok1* from *P. comata* and *Spok2*, *Spok3*, and *Spok4* form *P. anserina* (Grognet et al. 2014; Vogan et al. 2019). Despite all homologs showing a high degree of similarity, *Spok2*, *Spok3*, and *Spok4* show no epistatic interactions among themselves, operating independently during sexual crosses (Vogan et al. 2019). Conversely, *Spok1* provides resistance to all other homologs and can kill against *Spok2*, and *Spok3*, but not *Spok4* (Grognet et al. 2014; Vogan et al. 2019). The SPOK proteins appear to consist of a tripartite construction. The function of the first domain is unknown. The second domain has homology to nucleases and is responsible for the spore killing, while the third domain is a kinase and is required for resistance (Vogan et al. 2019). Site-directed mutagenesis of conserved sites has determined that the second domain (henceforth the "killing domain") possesses a lysine in the active site that is necessary for killing, while the third domain (the "resistance domain") has a canonical kinase active site with an aspartic acid residue at position 667 in *Spok3*. The functional separation of these two portions of the protein has been verified through generation of a truncated version that is lacking the entire resistance domain, which is lethal when expressed *in vitro* (Vogan et al. 2019). How resistance is conferred remains to be discovered, whereas the mechanism of killing appears to operate via general DNA degradation (Urquhart and Gardiner 2023). Of particular note is that this protein architecture, a nuclease domain associating with a kinase domain, is widely distributed across different domains of life (Zhang et al. 2016). It is proposed that this specific class of nuclease domain arose from bacterial transposons, later diversifying into various toxins which are involved with a range of biological conflicts (Zhang et al. 2016), including host-pathogen interactions (Van Damme et al. 2012) and gene drive systems (Lorenzen et al. 2008).

*Spok3* and *Spok4* are found at lower frequencies in *P. anserina* than *Spok1* or *Spok2* and are located at different genomic locations in different strains (Vogan et al. 2019). This is due to the fact that these two genes are situated within a giant transposable element, initially refereed to as the "*Spok* block", now renamed to *Enterprise* (Vogan et al. 2019, 2021). Along with a handful of other large transposons, *Enterprise* became a founding member of the superfamily of transposons called *Starships*. These TEs average around 110 kb in size and are noteworthy for the fact that they mobilize a large number of fungal genes both within and between species (Gluck-Thaler et al. 2022; Urquhart et al. 2023, 2024). Gene content analysis of the *Starships* has shown that *Enterprise* is not the only element to carry *Spok* genes, with numerous other elements harboring homologs (Gluck-Thaler et al. 2022). However, as no other *Spok* homologs have been

functionally validated to operate as meiotic drive genes and/or toxin/antidote genes, the role that these *Starship* associated *Spok* homologs play in fungal biology remains unknown.

To evaluate our hypothesis of the SPOK proteins' role in the maintenance and spread of ACs in *Fusarium*, we assessed the diversity of *Spok* homologs (henceforth *FuSpoks*) across the genus and their association with accessory genomic regions. We then determined whether a subset of these homologs can act as toxin/antidote proteins in a *S. cerevisiae* yeast system. Together, these results allow us to propose a model for the role of *FuSpoks* in the maintenance, origin, and spread of ACs in *Fusarium*.

## Materials and methods
### Phylogenetic analysis
We retrieved 146 high-quality RefSeq genome assemblies from 24 *Fusarium* species, selecting for assemblies that have been generated using long-read sequencing technologies (Supplementary Table 1). Genomes lacking annotations were annotated by liftover annotation using `Liftoff` (Shumate and Salzberg 2021), with the closest related reference genome with gene annotations used as the reference. This collection of *Fusarium* genome assemblies were stored in a `Mycotools` (Konkel and Slot 2023) database, which provided the framework for the identification of SPOK homologs and phylogenetic analysis. Such homologs were found using an HMM profile built from a collection of 250 protein sequences of the meiotic drive suppressor kinase protein family (IPR052396) present across *Fusarium*. The alignment of these reference amino acid sequences was created using `MAFFT` and `hmmbuild` was used to construct the final HMM model (Eddy 2011). Putative homologs were identified across this set of genomes using this HMM model and `hmmersearch`, with results based on a minimum e-value (0.001) and sequence similarity threshold (>30%) to the query. Protein sequences were generated based on these filtered hits and which represent full protein sequences based on existing genome annotations. However, coding sequence annotations of *Spok* genes are often incorrect and contain false introns. We retrieved the nucleotide sequence for each putative *Spok* gene and identified the existing reading frame, independently of existing gene annotations. In total, we recovered 437 putative *Spok* homologs from 14 *Fusarium* species (Supplementary Table 2), and retained 379 of these genes after removing duplicate genomes assemblies generated from the same *Fusarium* strains.

In addition to this set of SPOK homologs, we included a set of 33 manually curated SPOK protein sequences from *P. anserina* ($n = 4$), *F. oxysporum* ($n = 13$), *Fusarium poae* ($n = 15$), and *Fusarium vanettenii* ($n = 1$). This set of protein sequences were aligned using `MAFFT` and trimmed using `clipkit` (Steenwyk et al. 2020). We excluded 56 putative homologs prior the phylogenetic analysis if a homolog was i) missing catalytic core of resistance domain, ii) missing most or all of the coiled-coil domain, iii) the sequence similarity to any of the reference *Spok* sequences was too low (<30%). Using `IQ-tree` (Minh et al. 2020), substitution models were compared, and a maximum-likelihood phylogeny was constructed with ultrafast bootstrap approximation. The final phylogeny was visualized using `iTOL` (Letunic and Bork 2024).

### Repeated sequencing errors of *FuSpoks*
Because of their high similarity, we did not decide on any particular *FuSpoks* from *F. poae* strain DAOMC 252244 to amplify (this was also a result of the flanking regions being too similar to design unique primers). We were able to amplify *FPOAC1_003985* placed on chromosome 2, but consistently found an indel in the amplified sequence:

we found one more G (five rather than four) than the reference sequence at position 1,848 to 1,851. Looking at the nucleotide alignment to the other *FuSpoks* found in the genome, they all carry five rather than four Gs at this position. Note that the gene model on NCBI contains an intron that spans this region.

To investigate why this discrepancy existed, we downloaded the Nanopore and Illumina sequence reads for the BioProject PRJNA578270 from the SRA (SRR13483968 for the Nanopore data and SRR13023856 for the Illumina data), and aligned these to the published reference genome using Minimap2-2.24 (r1122) (Li 2018, 2021). For both sequence data sets we used samtools 1.14 (functions fixmate, sort and index with default options) (Danecek et al. 2021) followed by Picard tools MarkDuplicates 2.27.1 (Picard toolkit 2019) with options REMOVE_DUPLICATES=true and CREATE_INDEX=true. The resulting BAM files were visualized using IGV (Robinson et al. 2011; Thorvaldsdóttir et al. 2012; Robinson et al. 2017). In the Nanopore reads, the position was variably called with either four or five Gs in sequence. The Illumina reads had a drastic drop in coverage at this position, with no reads spanning the sequence of Gs. This made us draw the conclusion that the reference sequence is actually wrong, caused by a sequencing error in Nanopore. Likely, it is a combination of high GC content and repeats (Delahaye and Nicolas 2021).

The finding that the reference sequence was wrong about this sequence made us question a second (reportedly) variable site in the *FuSpok* sequences of this genome. About half of the copies predict a frameshift caused by a single base pair deletion of a G. We repeatedly found no Illumina reads covering the region, for the copies with the reported base pair deletion. In the Nanopore data, we found that these sites were variable, with around half of the reads having single rather than double Gs. There were also an inordinate amount of sequences covering the homologous site in the full copy sequence (with two Gs) on Contig_2. We thus concluded that the premature stop codon reported for these *FuSpoks* were technical errors during sequencing.

## Fungal strains and culturing conditions

For all experiments involving the expression of *Spok* genes in yeast, we used a modified version of the yeast strain BY4742, for which we replaced the *TRP1* gene by insertion of *URA3*. See *Construction of yeast strain* for details. Yeast were grown on Yeast Extract–Peptone–Dextrose (YEPD) media at 30 before transformation.

For amplification of a *Spok* homolog from *F. vanettenii* isolate 77-13-4, the strain was acquired from the Agricultural Research Service culture collection (NRRL), under the identifier NRRL 45880. The fungus was grown on potato dextrose agar at 27°C for 48 h, and the spores were collected by scraping the surface of the fungal growth with a flame sterilized loop and dipping it in an Eppendorf tube with 50 ml water. The spore solution was spread uniformly on a Potato dextrose agar (PDA) plate covered with cellophane film and incubated at 27°C for three days. Mycelia was then harvested by scraping off the cellophane and stored at −20°C for DNA extraction.

## Expression of *FuSpoks* in yeast
### Genes investigated

In addition to *Spok3* from *P. anserina*, we investigated the following genes: NECHADRAFT_82228, FOXG_14281, FOXG_14774, FOXG_07107, *FuSpok1*, *FuSpok2*.

### Construction of yeast strain

We replaced the *TRP1* gene in BY4742 (Winston et al. 1995; Baker Brachmann et al. 1998) by first amplifying *URA3* from plasmid DNA with primers that had 40 bp homology to the flanking regions of *TRP1* within the BY4742 genome, and 24 to 26 bp homology to *URA3*. The resulting PCR product was transformed into BY4742 using the lithium acetate/single-stranded carrier DNA/PEG method of transformation described by Gietz and Schiestl (2007), slightly modified to use overnight culture. Transformed culture was plated on synthetic dextrose (SD) medium lacking uracil (SD-U). Colonies were picked and streaked on SD-U and SD lacking tryptophan (-W), and one colony which grew on SD-U but on SD-W was chosen as our strain to use for the *Spok* expression assays.

### Vector construction

We extracted DNA from *F. vanettenii* (isolate 77-13-4) using the Quick-DNA™Fungal/Bacterial Microprep Kit (ZymoResearch) and confirmed fungal DNA presence with ITS1 and ITS4 primers. DNA from *F. poae* strain DAOMC 252244 was provided by Dr. David Overy, and DNA from *F. oxysporum* f. sp. *lycopersici* strain 4287 by Dr. Antonio Di Pietro. *Spok3* from *P. anserina* was amplified from previously obtained plasmids (Vogan et al. 2019).

The plasmid P001, a pRS413 derivative (Mumberg et al. 1995), includes a 2-micron origin, AmpR (for *E. coli*), TRP1 (for *S. cerevisiae*), and a galactose-responsive *pGAL1–pCYC1* hybrid promoter (Guarente et al. 1982) (Supplementary File 1). The *GFP* gene in P001 was replaced by *Spok* genes in our project.

High Fidelity Phusion polymerase (ThermoFisher) was used for PCR, except for two gap repair reactions where Q5 2x mastermix was used. Primers used are found in Supplementary Table 3. Primers were designed for restriction enzyme sites to amplify genes of interest from genomic DNA, targeting flanking regions where *FuSpok* sequences diverge. PCR products and the P001 backbone were digested, ligated, and transformed into 5-α Competent *E. coli* C2987H (New England Biolabs) for selection. Correct plasmid construction was confirmed by colony PCR and Sanger sequencing.

Purified plasmids (QIAGEN plasmid purification midi kit) were then used as templates for gap repair PCR, designed with long primers for precise integration into the P001 backbone. The digested plasmid and amplified gene were transformed into yeast using a lithium acetate/PEG method (Gietz and Schiestl 2007), and colonies were selected on SD-W agar plates. An exception to this approach was *FOXG_14774* from *F. oxysporum* f. sp. *lycopersici*, which was amplified directly from genomic DNA and used for gap repair.

### Site-directed mutagenesis

For the majority of the genes, we mutagenized the conserved aspartic acid residue into alanine through whole plasmid PCR of the constructed plasmids. The mutagenized plasmid was then cloned in *E. coli* and used as template for gap repair as described above. For *FOXG_07107* and *FOXG_14774* we instead made two separate PCR reactions for mutagenesis, producing one product for the first part of the gene (up until and extending 21 bp beyond the mutagenized base) and one product for the second part of the gene (starting 22 bp upstream of the mutagenized site and continuing to the end of the gene). These two products were combined with the plasmid backbone for a three-segment gap repair. For details of specific site mutations, see Supplementary Table 4.

### Growth assay

For assays of *Spok* toxicity, yeast containing the plasmids of interest were grown on SD-W for two days, after which single colonies

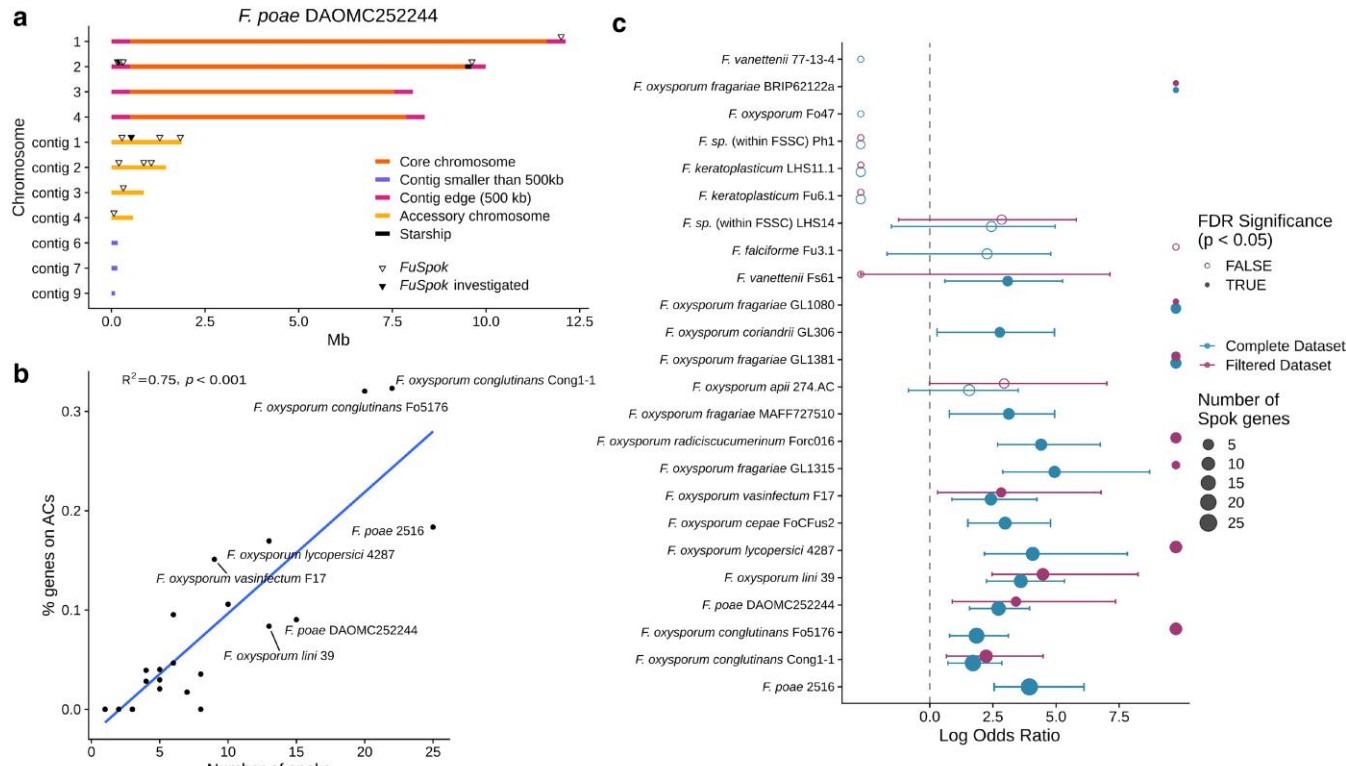

**Fig. 1.** a) Location of *FuSpoks* in *F. poae* strain DAOMC 252244. Chromosomes/contigs are colored based on their reported identity in Witte et al. (2021) or contig edges. Exceptions are contigs 6, 7, and 9, which are reported as ACs but are colored different because of their size (small contigs). b) There is a positive relation between the proportion of genes on accessory chromosomes and the number of *FuSpok* genes (within genomes for which accessory chromosome annotations were available, *n* = 24). c) *FuSpoks* are enriched in the accessory regions of *Fusarium* genomes. The odds ratio from Fisher's Exact test (with 95% confidence intervals) performed on the counts of *FuSpok* genes demonstrate an enrichment of *FuSpok* genes on accessory chromosomes in 16 of 24 genomes tested.

were picked into sterile water, serially diluted, and spotted onto SD-W with either 2% glucose (control, as genes are not expressed on this media) or galactose (on which the *pGAL1–pCYC1* promotor is induced and the genes are expressed). Plates where incubated at 30°C for either two days (glucose plates) or three days (galactose plates) before photos were taken.

## Results and discussion
### *FuSpoks* are rapidly diverging and enriched in accessory genomic regions

Casual observations have indicated that the homologs of the *Spok* genes in *Fusarium* are associated to the ACs and/or accessory regions of core chromosomes in some strains (Vogan et al. 2019). The location of *FuSpoks* in the *F. poae* strain DAOMC 252244 genome is given as an example (Fig. 1a). To determine if this is a general principle, and thus if there may be a inherent connection between these genomic features and the *FuSpoks*, we assembled a collection of 146 high-quality genomes from across the genus. In our previous research we had observed that it is difficult to accurately assemble *Spok* gene family sequences from short-read assemblies alone. This is likely due to a combination of high similarity among homologs, gene conversion events, and a propensity to be flanked by repetitive sequences, such as TEs (Vogan et al. 2019; Ament-Velásquez et al. 2022). Thus, to obtain an accurate assessment of the number and diversity of *FuSpoks*, we restricted our search to only genomes produced with long-read technologies. Unfortunately, this led to another unforeseen pitfall regarding genomes produced with Oxford Nanopore Technologies

(ONT). A number of *FuSpoks* were annotated as possessing introns in regions of the gene known to be critical for function, while the *Spok* genes in *Podospora* have no known introns in the coding region (Vogan et al. 2019). Manual inspection often revealed that the introns were predicted when stop codons were present in the sequence, suggesting that the genes with introns may in fact be pseudogenized. However, in some cases (particularly those involving *F. poae*), we found that while the ONT reads indicated that a stop codon was present in the gene, reads produced with Illumina technology did not. This is likely a result of repeatable errors in base calling that ONT is susceptible to (Delahaye and Nicolas 2021). As a result, we annotated protein sequences ourselves from all genomes, assuming no introns. Cases where stop codons were found prematurely were excluded from the final set, see Methods for specific details. This approach yielded 437 *FuSpok* homologs from 95 genomes (Supplemental Table 2). The genome with the highest number of *FuSpoks* was *F. poae* strain 2516 with 25 homologs, while many other strains were found to have none (Supplementary Table 1).

To assess whether *FuSpok* genes were specifically associated with ACs, we focused on a subset of 24 *Fusarium* genomes for which information on ACs were available, as *de novo* annotating accessory chromosomes across all 146 high-quality genomes was beyond the scope of this study. Firstly, we found a significant positive relationship between the percentage of genes found on accessory regions (used as a proxy for the relative size of the accessory genome) and the number of *FuSpoks* in each genome ($R = 0.75$, $P < 0.001$, Fig. 1b). Secondly, we conducted individual Fisher's Exact tests for each genome, comparing the number of

non-*FuSpok* genes to the number of *FuSpok* genes on accessory versus core regions, to test for enrichment. We found significant enrichment of *FuSpok* genes in the accessory regions in 16 of the 24 genomes tested (Fig. 1c). We also performed the same analyses on a dataset of *FuSpok* genes that were not present on small contigs (<500 kb), on *Starship* elements, or at the edges of scaffolds (500 kb), which are often enriched for accessory genomic content, but for which we cannot be certain are truly accessory given limitations of assembly completeness or insufficient annotation (Yang et al. 2020). This latter analysis changed the number of *FuSpoks* for 19 of the genomes, of which two changed from having *FuSpoks* significantly enriched in ACs to non-significant. While the utility of these statistical analyses demonstrate an association between *FuSpok* genes and ACs, they do not establish causation. We explore potential explanations for this distribution pattern in the section On the origin of accessory chromosomes in *Fusarium*.

To investigate patterns of gene family diversification of the *FuSpoks*, we generated a phylogeny of 379 *FuSpok* homologs from a set of genome assemblies with non-redudant taxonomic representatives ($n = 136$), as well as 29 *FuSpok* genes from our reference set (Fig. 2a). The topology of this phylogeny is generally consistent with existing *Spok* gene phylogenies (Grognet et al. 2014; Vogan et al. 2019), and can be divided into four main clades. *FuSpok* genes from *F. oxysporum* are prevalent across all clades, while homologs from the other species display sparser distributions, which may be due to the lower number of genomes available. One notable exception is *F. poae*, which exhibits a large number of homologs in clade IV. This is of particular interest as *F. poae* is closely related to *F. graminearum* and shares its general genome structure of having four large core chromosomes. However, unique to this species is the presence of multiple ACs (Witte et al. 2021), upon which many of the *FuSpoks* are located (Fig. 1a). Even more striking is the fact that we included 8 *F. graminearum* genomes in our analysis, none of which contained *FuSpok* homologs that passed our criteria (Supplementary Table 1). This provides strong circumstantial evidence that there is a link between the origin of ACs and *FuSpoks* within a given species/lineage.

As the most well represented *Fusarium* species, most *FuSpoks* were retrieved from *F. oxysporum* genomes. Many of these *FuSpoks* were very similar in terms of sequence and likely represent orthologs, *i.e.* the same gene in separate strains. As ACs are linked to the ability of specific lineages of *F. oxysporum* to infect specific plant hosts, we expected to see associations between *forma specialis* and the phylogenetic distribution of the *FuSpoks*. While there is a large amount of variation in copy number of the *FuSpoks* in specific assemblies (Supplementary Fig. 1; Supplementary Table 1), no other broad trends in the *FuSpok* phylogeny of *F. oxysporum* are apparent (Supplementary Fig. 2). However, more fine scaled analyses of AC synteny among the strains may be required to discern such patterns.

Given the incongruity of the *FuSpok* phylogeny with known relations among *Fusarium* species (Vogan et al. 2019), and the previous observations that the *Podospora Spok* genes are present on *Starships* (Vogan et al. 2021), we explored the association of these massive transposons with the *FuSpoks*. A total of 33 of the 408 *FuSpoks* in our phylogeny were found to be within known *Starships* (Fig. 2). The majority of the *FuSpoks* within *Starships* were found within *F. oxysporum* genomes, although some *Starship*-associated *Fuspoks* were also found within *F. solani F. poae*, *F. musae*, and *F. redolans*. The *Starships* containing *FuSpoks* are from *Starships* that are classified as different *Starship* families (Supplementary Table 2).

As the identification of *Starships* can be complicated by a number of factors (Gluck-Thaler and Vogan 2024), our annotations of these elements is likely incomplete. Nevertheless, our phylogeny of *Spok* homologs suggests that *FuSpoks* are rapidly gained and lost by *Starships* and that the horizontal transfer of *Starships* does influence *FuSpok* distributions. For example, the single *F. poae FuSpok* in clade III may have been transferred from *F. oxysporum* or *F. langsethiae*, although no *Starships* have been identified in the latter species (Fig. 2d). Given that the horizontal transfer of *Starships* has been linked to the outbreak of novel fungal diseases in crops (Bucknell et al. 2024; Urquhart et al. 2025), whether the presence of *FuSpoks* on *Starships* influences their transmission requires further attention.

## *Spok3* from *P. anserina* operates as a toxin and antidote in *S. cerevisiae*

Previously, we investigated the function of the *Spok* genes through genetic manipulations within *P. anserina*. While this provided numerous key insights to the functionality of the *Spok* genes, it had a number of drawbacks. In particular it was difficult to assay the toxin function of the genes as it was discovered that the *Spoks* not only kill developing sexual spores, but are active vegetatively. Thus, investigations of point mutants on the resistance function of the genes had to be inferred from a lack of transformants, rather than through observations of spore killing directly (Vogan et al. 2019). To remedy this issue, we developed a killing assay using the model yeast *S. cerevisiae*.

We utilised a vector based approach wherein the full length *Spok3* gene was placed under inducible expression via the *pGAL1*–*pCYC1* promoter. As compared to a control vector expressing GFP, a minor reduction in growth was observed, suggesting that the full length *Spok3* gene has a slight deleterious effect on yeast growth (Fig. 3). In *P. anserina*, a D677A point mutation, which is located within the active site of the resistance domain (Vogan et al. 2019), was shown to reduce the resistance function of the protein. Congruently, this same point mutation reduced yeast growth substantially (Fig. 3). This result indicates that the *Spok3* gene is an active toxin and antidote within *S. cerevisiae*. This agrees with previous results by Urquhart and Gardiner (2023) who obtained similar results with assays performed with the *Spok1* gene from *P. comata*. We confirmed that two additional modifications of *Spok3* operate similarly in the yeast assay as in *P. anserina*, namely a truncated version of *Spok3*, which has a stop codon inserted after amino acid 490, and the K240A point mutation in this truncated version. The truncation removes the entire resistance domain, while the point mutation is in the active site of the killing domain and disables the toxicity function of the gene (Supplementary Fig. 3). Strikingly, this truncated version of *Spok3* inhibited yeast growth to a much stronger extent than the D667A point mutation, suggesting that some resistance activity is still maintained in the mutated active site.

## *FuSpok* genes function as toxin/antidote proteins

Although the *Spok* homologs distributed throughout *Fusarium* show a high degree of similarity to the *Podospora* genes, including conservation of characterized catalytic amino acids (Supplementary File 2), as yet it is unknown whether they possess toxin and/or antidote function. To address this, we selected six homologs from three different species to investigate with the yeast assay (Table 1). We chose genes based on a combination of factors, including phylogentic diversity and genomic location. We chose three genes from the well-studied tomato infecting strain *F. oxysporum* f. sp. *lycopersici* 4287. Gene *FOXG_14281* and

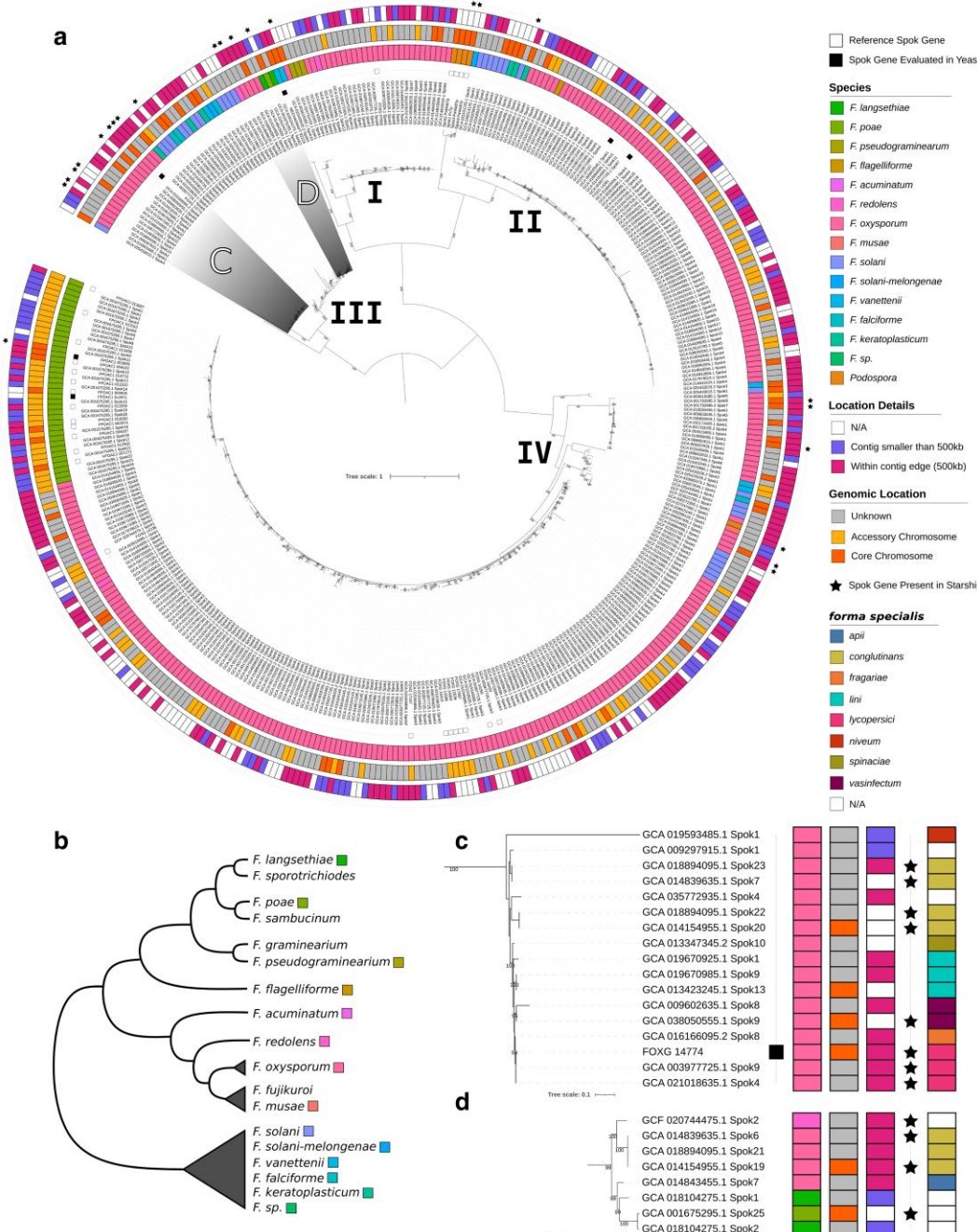

**Fig. 2.** a) Maximum-likelihood phylogeny (mid-point rooted) of *FuSpok* gene homologs. The support values are ultrafast bootstrap (*n* = 1,000) approximations from IQ-tree, with only bootstraps with a value of > = 95 shown. Tips annotated with a black square indicate *Spok* genes that are part of the reference set (Supplementary Table 2), with filled squares denoting the *FuSpok* genes which were assayed in this study. The innermost colored ring of annotations denotes species identity of the *Fusarium* genomes within the phylogeny. The middle colored ring contains annotations of the genomic locations of *FuSpok* genes that could be assigned to accessory or core chromosomes. The outer-most colored ring show the genomic organization of *FuSpok* genes that are found at contig edges or on small contigs of their respective genome assemblies. Tips annotated with stars denote *FuSpoks* which were found inside of *Starship* elements. b) A schematic phylogeny of *Fusarium* species relationships is included for reference. Relevant species complexes are indicated with triangles. Panels c) and d) depict subtrees enriched in *FuSpok* genes present within *Starships*. Additional colored annotations demonstrate the *forma specialis* designations for *F. oxysporum* genomes. *F. sp* refers to strains present within the *Fusarium solani* species complex that were not assigned to a specific species.

*FOXG_07107* are both located on ACs. Furthermore, gene *FOXG_14281* is located on chromosome 14 which was shown to transfer between strains in Ma et al. (2010) and retained in heterokaryotic strains experiencing genome reduction in (Shahi et al. 2016). Gene *FOXG_14774* is residing within the *Starship Arwing* (Gluck-Thaler et al. 2022; Gluck-Thaler and Vogan 2024) on chromosome 12. Additionally, we selected the

*NECHADRAFT_82228* (accession: XM_003041460.1) gene from *F. vanettenii* as it was investigated previously and shown to possess toxin activity, but no antidote activity (Grognet et al. 2014).

We were able to amplify two genes from the species *F. poae*, one located on a core chromosomes (*FPOAC1_003985* on chromosome 2), and the other on an accessory contig (*FPOAC1_012971* on contig 1). We did not *a priori* chose these genes, as the multiple highly

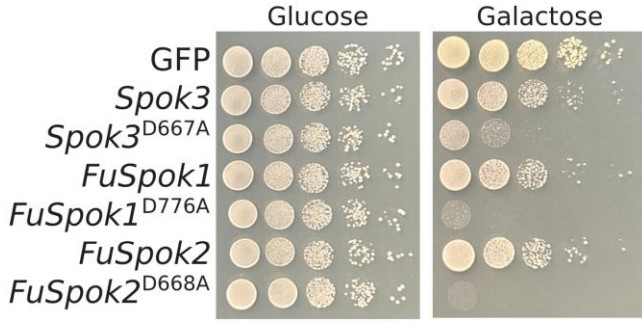

**Fig. 3.** Mutated copies of three *Spok* genes kill in *S. cerevisiae*. Photos show spots of yeast growth after two (Glucose) or three days (Galactose) at 30. Left panel (Glucose) is the control and shows yeast carrying, but not expressing, each gene noted on the left of the photo. The right panel (Galactose) shows yeast growth when expressing each gene.

similar copies made design of specific primers difficult. We did however chose to amplify genes from the particular *F. poae* DAOMC 252244 because of two reasons. First, there are a large number of *FuSpoks* in this strain, many of which are highly similar, suggesting recent expansion. Second, the presence of ACs is unique in this species among close relatives. When copies of the *FuSpoks* from *F. poae* that possess point mutations in the resistance domain were introduced to *S. cerevisiae*, a strong inhibition of growth was observed, greater even than what was obtained with *Spok3* (Fig. 3). Given their function, we name these genes as *FuSpok1* for FPOAC1_003985 and *FuSpok2* for FPOAC1_012971, see Table 1. This result provides strong support for a role of *FuSpoks* in the maintenance and/or spread of accessory chromosomes in *Fusarium*. In conjunction with the results from the phylogeny, we can infer a scenario where the acquisition of a *FuSpok* gene from *F. oxysporum* provided the means for the evolution of ACs in *F. poae*. Additionally, given the large number of highly similar homologs in clade IV from other *Fusarium* species, it suggests that hundreds of *FuSpoks* may be functional toxin/antidote genes (Fig. 2). Thus supporting our main hypothesis that the *FuSpoks* are key players in the evolution of ACs across the genus.

Results from the other four investigated genes (Table 1) showed no impact on yeast growth, including *NECHADRAFT_82228* (Supplementary Fig. 4). A number of factors may be responsible for these observations. First, despite conservation of known active sites, these genes may no longer function due to changes elsewhere in the proteins. This is partially supported by the fact that *NECHADRAFT_82228* displayed no resistance function in previous experiments, despite appearing to have a fully functional domain (Grognet et al. 2014). However, the fact that we did not observe inhibition in yeast growth, even of the unmutated copy, while previous experiments demonstrated toxic properties in *P. anserina*, may indicate that our assay is not sensitive enough to capture the whole spectrum of toxin effects. We see variation in the extent of yeast growth inhibition between *Spok3* and the *F. poae* homologs, implying that not all proteins impact *S. cerevisiae* to the same degree. It is therefore possible that the NECHADRAFT_82228 is a much weaker toxin, and that its effect is not observable under the assayed conditions. Urquhart and Gardiner (2023) observed increased sensitivity to *Spok1* when a DNA repair mutant was used, which could be a viable option for future assays of *FuSpok* genes. Additionally, as the truncated *Spok3* exhibited a much stronger toxic effect than the point mutation, it is possible that mutating the aspartic acid residue has minimal impact on resistance function for some of the homologs. Lastly, proteins may not be expressed or produced correctly from

the constructed vectors for unknown reasons. Alternatively, it may be the case that the *FuSpoks* do not generally operate as toxin/antidote genes, but that their native function has been co-opted for selfish behavior separately in both *Podospora* and *F. poae*. However, we find this explanation unlikely given the degree of gene family expansion observed in strains like *F. oxysporum* f. sp. *lycopersici* 4287.

## Model for the maintenance of accessory chromosomes by *FuSpok* genes in *Fusarium*

Given the different points of evidence presented herein, we can now propose a model for the role of the *FuSpok* genes in *Fusarium*. Transcriptional evidence indicates that the *Spok* genes are consistently expressed in *P. anserina*, including in vegetative growth, suggesting that the entire mycelia is infused with SPOK protein (Vogan et al. 2019). To avoid the toxic effects of the protein, a nucleus must either possess a copy of the specific *Spok* gene or share a cell with a nucleus that does (Grognet et al. 2014; Vogan et al. 2019). One of the greater mysteries surrounding the *Spok* genes is how a single protein is capable of acting as both a toxin and an antidote. In many other systems the two roles are encoded either by separate, linked genes, or by alternate isoforms of the same gene (Bravo Núñez et al. 2018). Although the mechanism is unknown, it follows that there should be two separate forms of the SPOK protein, one responsible for toxicity, and the other for the antidote function. In systems where separate isoforms of the same gene encode the two functions, the antidote isoform is often shorter-lived than the toxin. For example, this is how spore killing is enacted in the model yeast *Schizosaccharomyces pombe* (Hu et al. 2017; Nuckolls et al. 2017). This is necessary in order to create an asymmetry whereby cells with the toxin/antidote system are protected from self-toxicity, while those cells that do not inherit the gene(s) are killed. Previous work by van der Gaag (2005) supports this model of asymmetric stability for the *Spok* genes in *P. anserina*. As such, if a nucleus were to lose an AC during growth— either through non-disjunction at mitosis or other means—and become isolated in its own cell, it will lack the antidote function. Since the nucleus would have been exposed to the *Spok* product during growth, as the antidote function degrades the nucleus will experience the toxic effects, *i.e.* DNA damage, and break down (Fig. 4). This resembles the well-described type II toxin–antitoxin systems in bacteria where a stable toxic protein is being neutralized by a more unstable antitoxin that requires constant production. The toxin–antitoxin systems in bacteria not only aid in the maintenance of the plasmid, but also of genomic pathogenicity islands, integrons, or entire chromosomes (Qiu et al. 2022).

So far, the understanding has been that SPOK proteins in *Podospora* only influenced transmission during meiosis, but our updated model suggests that they act throughout the entire life cycle to bias transmission. This aspect is especially relevant when considering genomic reductions after parasexual fusion, which is a likely cause of observed horizontal chromosome transfer in *F. oxysporum* (Vlaardingerbroek et al. 2016). In *Fusarium*, this mitotic drive has significant implications, as *FuSpoks* are located on accessory chromosomes that carry genes that increase pathogenicity of the fungus. Understanding that these chromosomes are not only selected for during host infection, but are also actively transmitted and propagated within populations provides valuable insights for predicting the spread of these pathogen-causing elements.

It is important to note that our model assumes a single functional SPOK protein operating in an otherwise empty genome. In reality, as we have shown above (Fig. 1), this is a rare scenario. Rather, out of the 95 genomes we found harboring *FuSpoks*, 63

**Table 1.** *FuSpoks* which toxicity we evaluated in yeast.

| Species | Strain | Locus Tag | Gene Name | Result in yeast | GenBank accession |
|---|---|---|---|---|---|
| *Fusarium vanettenii* | NRRL 45880 | NECHADRAFT_82228[a] | Unnamed | No effect | NA |
| *Fusarium oxysporum* f. sp. *lycopersici* | Fol4287 | FOXG_14281 | Unnamed | No effect | NA |
| *Fusarium oxysporum* f. sp. *lycopersici* | Fol4287 | FOXG_14774 | Unnamed | No effect | NA |
| *Fusarium oxysporum* f. sp. *lycopersici* | Fol4287 | FOXG_07107 | Unnamed | No effect | NA |
| *Fusarium poae* | DAOMC 252244 | FPOAC1_003985 | *FuSpok1* | Toxicity | PQ540985 |
| *Fusarium poae* | DAOMC 252244 | FPOAC1_012971 | *FuSpok2* | Toxicity | PQ540984 |

The Result in yeast column refers to the effect of the protein with a point mutation leading to the replacement of a predicted aspartic acid residue with an alanine (corresponding to the D667A mutation previously investigated for SPOK3). [a]NCBI gene model includes putative false intron

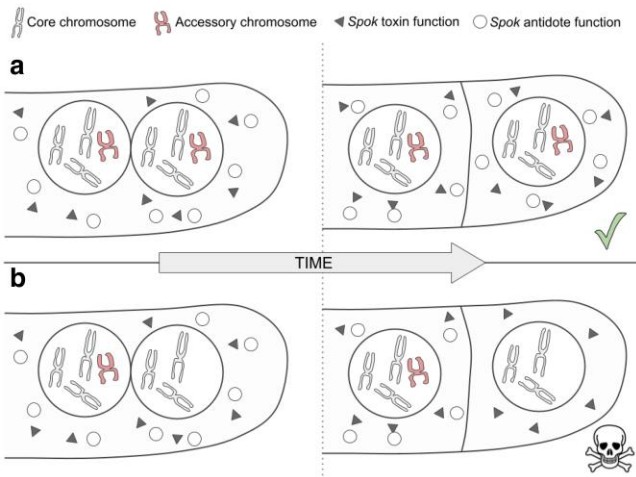

**Fig. 4.** Model for the maintenance of accessory chromosomes by *FuSpoks*. Triangles represent the toxin function of the proteins, and circles represent the antidote function. Core chromosomes are depicted in grey, with ACs in red. a) A dividing cell within a hyphal tip, expressing a SPOK protein from an AC. b) A dividing cell within a hyphal tip, expressing a SPOK protein from an AC. The AC is lost during mitotic division, resulting in a cell containing *Spok* toxin, but with no ability to produce *Spok* antidote. This cell is expected to die.

carried more than one such gene (Supplementary Table 1). Notably, these results are for genes that passed our rather stringent filtering criteria, verifying that they were indeed full copies. Pseudogenized copies are therefore not presented, but exist in many genomes. If and how selection acts to maintain a functional toxin and antidote domain when there are other, sometimes identical, copies of the gene present in the genome, is not evident. In *P. anserina*, the three *Spok* genes are highly similar, yet have functionally diverged to act as independent toxin/antidote genes (Vogan et al. 2019). It is thus possible that all *FuSpoks* in a given genome represent unique toxin/antidotes. Furthermore, there is evidence from *P. anserina* that versions of *Spok* genes exist with functional antidote domains, but non-functional toxin domains (Grognet et al. 2014). Antidote only genes are also predicted in the rapidly evolving *wtf* gene family in *S. pombe* (Eickbush et al. 2019). The *wtf* family, similar to our *Spok* family, show copy number variations between isolates, amino acid substitutions, and variable size of DNA sequence repeats (in our case the coiled-coil domain) (Eickbush et al. 2019). It should not be unexpected to find *FuSpoks* on core chromosomes (Fig. 2), nor is it unexpected to find non-functional or weakly functional copies, given their rapid expansion and diversification. Rather, these aspects reflect the complex intragenomic conflict among the many copies of these genes, ACs, and the core genome. Deeper investigations into individual species or strains should be undertaken to fully appreciate the impact of the *FuSpoks* on their host.

## On the origin of accessory chromosomes in *Fusarium*

ACs are common across fungi, but their origins remain a mystery. We propose a model for how the presence of *Spok* genes could help solve this riddle. Assume that a novel chromosome is formed, possibly via chromosomal fission or other structural rearrangements, in a genomic region containing a *Spok* gene. Then, the selfish properties of the *Spok* gene would offer a critical stabilizing mechanism so that the newly formed chromosome (possibly operating with a suboptimal neocentromere) would not be lost. Alternatively, *Spok* genes could enter from outside the genome, for example via transposition of a *Starship*. The fact that *Spoks* are frequently found on *Starship* transposons (Gluck-Thaler et al. 2022) suggest this as a possible route into the genome. As *Starships* often degrade, sometimes rapidly (Urquhart et al. 2024), this process can be difficult to observe, but the fact that at least some *Spok* genes studied here are on *Starships* supports such a role. In contrast to previous models on the origin and maintenance of ACs such as them carrying genes beneficial to the host, our model does not require specific environmental conditions or growth conditions to be met for selection to start favoring these *Spok*-carrying chromosomes. If a *Spok*-carrying AC subsequently becomes beneficial due to the movement and/or *de novo* creation of genes with roles in rapidly evolving traits, like virulence, on said chromosome, the *Spok* genes may no longer be needed to maintain it, and could be lost or transferred to other genomic regions. Thus, while we find *Spok* genes to be enriched on ACs, this would explain why they are not a strict requirement for their existence. Other selfish properties of ACs, such as the meiotic drive observed in *Z. tritici* (Fouché et al. 2018), may also play a role in the existence of ACs that do not carry *Spok* genes.

Homologs of *Spok* genes are not restricted to *Fusarium* and *Podospora*, but are found distributed throughout a broad range of species across the Pezizomycotina. Considering *Starships* can mobilize both within and between species (Urquhart et al. 2023, 2024), their movement likely contributes to the incongruous phylogenetic distribution of *Spok* genes (Vogan et al. 2019). It remains to be answered whether *Starships* and *Spok* genes have co-evolved since their respective origins, or if *Spoks* are recurrently found in *Starships* by chance. In either case, as the presence of *Spok* genes does not coincide directly with the presence of ACs, other factors must be in play. Nevertheless, understanding that *Spok* genes play a key role in AC evolution is an important step in determining why ACs appear in some species and not others.

## Conclusions

We have provided evidence that the role of *Spok* homologs in *Fusarium* could be to maintain ACs during vegetative growth. Due to the large number of *FuSpoks* in given strains of *Fusarium*

and the fact that there can be epistatic interactions between different homologs (Grognet et al. 2014), it is difficult to predict whether every *FuSpok* plays an active role in accessory genome maintenance. The *Spok* genes could also play a role in the parasexual cycle, which is key for the process of whole chromosome transfer in *Fusarium*, and likely plays a role in the evolution of virulence (Shahi et al. 2016). Understanding if different *FuSpoks* in a given genome have epistatic interactions and thus dictate which ACs can be transferred between strains should be a priority of future work. Although the large number and diversity of these genes makes this a formidable challenge. Additionally, follow up work to demonstrate the role of *FuSpoks* within *Fusarium* itself is required to verify the model proposed here. This work is an important first step in deciphering the role of selfish genomic elements in the origin, maintenance, and spread of ACs and thus of virulence in devastating plant pathogens with far reaching implications to managing these important fungi for agriculture and society more broadly.

## Data availability

Strains and plasmids are available upon request. Supplementary Table 3 contains primers used. Supplemental material available at GENETICS online.

## Acknowledgments

Dr. David Overy and Dr. Thomas Witte from Agriculture and AgriFood Canada provided DNA and sequence data from *F. poae* strain DAOMC 252244. Dr. Antonio Di Pietro from University of Córdoba, Spain, provided DNA from *Fusarium oxysporum* f. sp. *lycopersici* strain 4287. Thank you to the USDA ARS Culture Collection for providing strain NRRL 45880. Thanks to Armin Rassooli Tilehnovi for important work on exploring *Fusarium Spok* homolog diveristy.

## Funding

L.S. was funded by Carl Tryggers Foundation project CTS 20:477. A.A.V. was funded by FORMAS 2019-01227 and the Swedish research council 2021-04290. A.M. was funded by an Erasmus trainee scholarship and a Study scholarship from the C.M. Lerici Foundation. A.U. was funded by a Wenner-Gren Foundation postdoctoral scholarship.

Conflicts of interest. None declared.

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
