## [Peer Review File · Genetics]

The role of toxin/antidote genes in the maintenance and evolution of accessory chromosomes in *Fusarium*

Linnea Sandell, Adrian Forsythe, Anna Mirandola, Samuel Jorayev, Andrew Urquhart, Alexandra Granger-Farbos, Sven Saupe, and Aaron Vogan

NOTE: The reviews and decision letters are unedited and appear as submitted by the reviewers.

In extremely rare instances and as determined by a Senior Editor or the EIC, portions of a review may be redacted. If a review is signed, the reviewer has agreed to no longer remain anonymous.

The review history appears in chronological order.

Review Timeline:

Submission Date:	2025-04-11
Editorial Decision:	2025-05-22
Resubmission Received:	2025-08-26
Accepted:	2025-09-10

May 22, 2025

GENETICS-2025-308066

The role of toxin/antidote genes in the maintenance and evolution of accessory chromosomes in *Fusarium*

Dear Dr. Vogan:

Two experts in the field have reviewed your manuscript, and I have read it as well. I am pleased to inform you that, with minor revisions, it is potentially suitable for publication in GENETICS. The reviewers have comments and concerns that need to be addressed in a revised manuscript. You can read their reviews at the end of this email.

It is most important that you address the following in your resubmission:

The Reviewers have indicated several areas to strengthen the manuscript and provide some context that might temper the conclusions, specifically that they raise question that while there is statistical support for the FuSpok gene and AC relationships, there are only a limited number of genomes showing this pattern. There is concern that there may not be enough examples so that either some comments on the limitations of the current sample size or provide other context to the strength of these interpretations in the conclusions.

There was also a concern raised about the rationale for inclusion of the specific hypothetical genes and the multiple growth media that needs some more justification or context.

The integration/comparison to the *Podospora* results I imagine are connected based on previous work in this field and from your labs but the transition to this inclusion or context needs more description or linkage.

We look forward to receiving your revised manuscript. Please let the editorial office know approximately how long you expect to need for revisions.

Upon resubmission, please include:

1. A clean version of your manuscript;
2. A marked version of your manuscript in which you highlight significant revisions carried out in response to the major points raised by the editor/reviewers (track changes is acceptable if preferred);
3. A detailed response to the editor's/reviewers' comments and to the concerns listed above. Please reference line numbers in this response to aid the editors.

Additionally, please ensure that your resubmission is formatted for GENETICS.

<https://academic.oup.com/genetics/pages/general-instructions>

Follow this link to submit the revised manuscript: Link Not Available

Sincerely,

Jason Stajich
Associate Editor
GENETICS

Approved by:
David Begun
Senior Editor
GENETICS

Reviewer #1 :

This paper presents a comprehensive overview and phylogenetic analysis of Spoks, a class of toxin/antidote genes in fungi, in the fungal genus *Fusarium*. Enrichment in accessory regions is shown as well as presence in 'Starships' (giant transposons) for a subset of Spok genes. A screen in yeast demonstrated toxicity of two of the six tested Spok genes (in which the antidote part was mutated). A model is presented for how Spok genes can stabilize accessory chromosomes by killing cellular compartments

without them.

This is a well written paper about a very exciting topic. I have the following suggestions for improvement.

1. Concerning the concept of causality. Mention is made of a possible 'causal connection' between accessory chromosomes/regions and FuSpoks (p. 4 line 50) and, after showing enrichment of Spoks in accessory regions, it is stated that "it does not establish causality". I would like to see some clarification of what could be causing what and what the nature of this causality is. I am presuming what is meant is causality in terms of selection, not a mechanistic causality. So Spoks would not 'cause' accessory chromosomes/regions to emerge but can act to maintain them in evolution. In relation to this, on page 9 line 1-2 it is stated: "Selection may then lead to the movement and/or de novo creation of genes with roles in ..." which is a bit confusing because selection does not 'lead to movement and/or de novo creation' but only acts to maintain 'moved' or 'de novo created' genetic elements.
2. Concerning the concept of orthologs. It is stated that "Many of these FuSpoks were very similar in sequence similarity [*see below] and likely represent orthologs (p. 5 line 4). I am not sure if the definition of ortholog is correctly applied here. In my understanding, 'orthology' denotes a relationship of descent between genes in different species, from the same ancestral gene in the last common ancestor, implied by synteny. Sequence similarity is not a primary measure of orthology, and gene expansions make orthology quite hard to assess.
3. Accessory versus contig edge (Fig. 1). I understand that an arbitrary definition of 'contig edge' of 500 kb was used but now contigs 7 and 9 seem to be only 'contig edge' and not 'accessory' which gives a false impression. I do not see a solution but perhaps it suffices to state this in the text and/or figure legend.
4. FuSpoks may have been introduced to *F. poae* by a starship from *F. oxysporum* Starship (p. 5 line 27-28). Similarity between the respective starships would make this plausible. Is that the case?
5. The legend in Fig. 2 mentions a 'highlight' for the clade containing the majority of *F. poae* FuSpok genes: I did not notice this highlight in the figure.
6. Concerning Figure 3: It would be better in my view if the title of this figure states the conclusion of the experiment. A legend should be added explaining the experiment.
7. The section on temperature and spore killing seems unnecessary. It could just be mentioned in the Discussion if at all necessary and presented in a separate paper.
8. Textual issues:
 - Amounts > numbers? (p. 1 line 3)
 - Benign media (p. 1 line 41) > is this a correct term?
 - Phylogeny 'separated' > 'divided' (p. 4 lines 105 and 107)
 - Remove comma's in p. 4 line 117 and p. 9 line 38
 - P. 5 line 3: delete 'similarity'
 - P. 7 line 68: Temperature effects > affects?
 - P. 9 line 21: directly > strictly?

Reviewer #2 :

Comments for the Author:

1. What is the functional and biological relevance of the hypothetical genes NECHADRAFT_82228, FOXG_14281, FOXG_14774, FOXG_07107 used in the study?
2. In the growth assay, both glucose and galactose media were used. However, the rationale for using two types of media is unclear. Since the glucose data do not appear to add any relevant insight, consider whether those results could be omitted or more clearly justified.
3. Out of the 146 genomes analysed, what proportion lacked any FuSpok genes and how does it impact the overall interpretation of the hypothesis?
The study states that not all Fusarium strains contain FuSpok genes and focuses analysis on 18 genomes, of which only 10 show FuSpok genes on accessory chromosomes (ACs). Given this limited sample, how robust is the claim regarding the association of FuSpok genes with ACs?
4. Line 98 of the Results and Discussion section implies that the analysis was conducted to explore functional divergence of Spok genes within lineages. However, the following paragraph focuses more on differences in distribution across species rather than on functional divergence. Consider rephrasing the opening line for accuracy.
5. The statement that "The effect of decreased temperature on Spok2 and Spok3 suggests that this condition stabilizes the

antidote function for long enough that the spores survive before both toxin and antidote function eventually degrade" is speculative. Please provide a reference or clarify that this is just an assumption.

Reviewer #3 :

The role of toxin/antidote genes in the maintenance and evolution of accessory chromosomes in *Fusarium*

The authors bring together two fascinating areas of genome biology, (1) gene drive, mediated by spore killing genes, Spok genes, and (2) accessory chromosomes. Fungi are an important group with which to study these processes because of the range of genome conformations they maintain, combined with their ability to invade new environments. All of this work is set against a backdrop of Starship TEs.

This is an exciting frontier of biology where little is known and we think the authors bring together some nice evidence for their model on the maintenance of Spok genes, usually implicated in spore killing, providing here a novel mechanism for their maintenance during vegetative growth. Being such a novel area of biology, this model is an important step providing something for other researchers to test, confirm/reject. However, some uncertainty in our ability to decipher figure 2, and the use of language linking the maintenance of Spok genes specifically with accessory chromosomes, which perhaps overstates causality, needs further consideration. This is in part due to the number of analyses/datasets included. The authors also introduce work from *P. anserina* as well as *Fusarium*, with temperature experiments and phenotypes in *Saccharomyces cerevisiae* - this could have been several papers, split in a number of different ways. Our comments focus on a clarification of the relationship between genome quality (ability to assemble ACs and the observations based on those ACs), clarifications around figure 2, as well as over assertion of the intrinsic role of Spoks for maintenance of accessory chromosomes, as they are absent in some of them.

Genome quality

To what extent is the observation of Spok gene presence/density in core and accessory chromosomes impacted by the quality of the genomes in the set? There was a reasonable amount of filtering from 146 genomes down to 94, and in figure 1 just 18 species were used. Across these 94 genomes there was presumably varying degrees of completeness. One would expect that completeness has a disproportionate effect on the quality of ACs. Does the Spok presence / density between core and accessory content relate to genome quality? For example, the authors find that 3/4 of Spoks are in ACs but to what extent do the authors have complete AC assemblies across their tree? What are the relative densities of Spok genes only on the core genomes, and is there a relationship between those densities, and the genomes for which they have well assembled ACs?

Figure 2

Figure 2 contains a great deal of information, and we spent some time referring to it to confirm statements in the MS. Being such a key figure, it is important that it is accurate and a good deal of our uncertainty about the work could be cleared up by addressing suggestions on this figure. For instance, the statement 'the single *F. poae* FuSpok in clade III may have been transferred from *Fusarium langsethiae*' (p5 lines 38-40)' is currently hard for the reader to assess.

Firstly, we suggest reconsidering using colour for three separate categories, especially with high similarity of colours within and across legends.

Secondly, the incongruence of the tree compared with species relationships is not clear for readers who are not intimately familiar with *Fusarium* taxonomy. It would be informative, for instance, for there to be some way for the reader to cross-check *Fusarium* species relationships with the relationships in the Spok tree. This could be represented as a species tree shown alongside the *Fusarium* legend. A visualisation of species relationships would also strengthen certain parts of the text e.g. where the difference in AC and Spok content in *F. poae* and *F. graminearum* is discussed on p4 lines 110-141; the authors may also want to cite Armer et al. 2024 (<https://doi.org/10.1016/j.funbio.2024.07.004>) to illustrate this latter point.

Finally, there are also some things that need to be clarified or corrected in the figure 2 legend. We assume that 'FSSC' refers to the *F. solani* species complex (syn. *Neocosmospora*), however FSSC species are also listed separately (*F. solani*, *F. falciforme*, *F. vanettenii*, *F. keratoplasticum*, *F. solani-melongenae*), so it is not clear what this category refers to. This is also the case in supplementary figure 1. *F. oxysporum* also appears twice in the legend, and there is an 'unknown' category in the genomic location legend which we cannot see being used in the figure itself.

P5ln22 main text states that ...'Spok genes from different Starship families present in different *Fusarium* species'. How can we see in figure 2 that these are from different Starship families? Or where can we see this information?

Assertive language

The barrier of proof to causation is high and, while we agree that Spoks have a role in the maintenance and evolution of accessory chromosomes in *Fusarium* (in the title) the language used throughout overstates this role. We list cases for consideration below.

The authors have provided evidence that Spoks are enriched on ACs in some cases, however we question whether 10 out of 18 genomes (p4 line 91/figure 1A) is a convincing number to conclude that Spok genes are a predominant cause of AC persistence. Evidently ACs are maintained for other reasons in closely related genomes - incidentally it would be nice to order the y axis by taxonomic relationships in figure 1A. That this is not a universal mechanism is also suggested by the fact that 2 of the 6 Spoks which were evaluated in yeast were located on ACs in *F. oxysporum* but were not found to have toxicity.

P1In10: 'Finally, as Spok genes are mobilized by the newly described TE superfamily Starships, it suggests these TEs play crucial roles in forming accessory chromosomes and regions'. This is quite a jump from Spoks in TEs to TEs in the formation of accessory chromosomes. This paper is not about the formation of accessory chromosomes.

P8In88 'but the fact that at least some Spok genes studied here are on Starships supports such a role'. Only very weakly. To do this analysis properly one would have to separate the signal from Spoks both inside and outside Starships, from that signal of the hundreds of other genes both inside and outside Starships.

The section 'On the origin of accessory chromomes in Fusarium' is a little fanciful and we would suggest a rewrite that considers more of the forces that create and maintain ACs. Perhaps even venturing into sex chromosomes, repeats and their maintenance. The authors state that a chromosomal fission could produce an accessory chromosome. Presumably the other important genes that already happen to be on that region, would also play a role in its preservation? This whole section is written without much if any attention to anything other than the impact of Spoks on genome structure. For example, p8In81:

'Alternatively, since Spok genes are frequently found on Starship transposons (Gluck-Thaler et al. 2022), a Spok gene could be brought to a novel chromosome soon after formation via transposition of a Starship, thus precluding the need for Spok genes in the background or core genome'. This argument should be backed up by evidence that Spok incorporation via HGT is more likely to occur in an AC than a core chromosome. Where, if incorporated into a core chromosome, the Spok would have a guaranteed its chance of being transmitted vertically, without the additional requirement to now have to also ensure the transmission of an AC.

P9In1 After Spoks are present in and maintaining the AC 'selection may then lead to the movement and/or de novo creation of genes with roles in rapidly evolving traits, like virulence, on said chromosome'. Why wouldn't any of these non-selfish processes already have occurred on the AC, and be maintaining them in their own right? This highlights the authors drive to put Spoks at the centre of every argument on the generation and maintenance of ACs. It reduces trust in objectivity.

P9In17: 'It remains to be answered whether there is a deep evolutionary link between the Starships and Spok genes or if Starships recurrently abduct Spoks'. A Starship cannot abduct anything.

Minor comments

P. anserina work comes quite out of the blue, should this have another header?

Spore killing reduced at 22 and returned at 27. Can the authors add something in the introduction to account for the life history and environment these fungi are exposed to. These temperatures seemed very high and had us questioning how regularly these fungi are exposed to these high temperatures, at which Spoks have any spore killing ability.

Supplementary tables are all missing - the captions are there but no tables

P4In52-76 - this bit feels more like pre-processing/methods than results?

Associate Editor Comments:

Summary and Overall Critique:

The study titled "The role of toxin/antidote genes in the maintenance and evolution of accessory chromosomes in *Fusarium*" is well designed investigation to analyse genomic localisation, phylogentic distribution and potential mobility of spore killing (*FuSpok*) genes. The study explicitly explores how *FuSpok* genes may contribute to the maintenance and spread of accessory chromosomes in the *Fusarium* genus.

- The study discusses four *Spok* genes. *Spok1* is resistant to all others, and it kills *Spok2* and *Spok3*, but not *Spok4*.
- The killing domain of *Spok* protein, the second domain, possesses a lysine in the active site that is necessary for killing.
- The third domain or resistance/antidote domain, has an aspartic acid at the active site, position 667, in *Spok3*.
- The authors make an important comparison between sequencing errors observed in Oxford Nanopore Technologies (ONT) and Illumina sequencing, noting limitations of ONT. They went to additional effort to manually annotate protein sequences across all genomes to ensure accuracy.
- The effect of temperature on the killing function of *Spok* genes was assessed between 22–25 °C. The study concludes that either *Spok2*, *Spok3*, or both lose their killing ability at 22 °C.
- Using a binomial test, the authors demonstrated a significant association between *FuSpok* genes and accessory chromosomes.

Comments for the Author:

1. What is the functional and biological relevance of the hypothetical genes NECHADRAFT_82228, FOXG_14281, FOXG_14774, FOXG_07107 used in the study?
2. In the growth assay, both glucose and galactose media were used. However, the rationale for using two types of media is unclear. Since the glucose data do not appear to add any relevant insight, consider whether those results could be omitted or more clearly justified.
3. Out of the 146 genomes analysed, what proportion lacked any *FuSpok* genes and how does it impact the overall interpretation of the hypothesis?

The study states that not all *Fusarium* strains contain *FuSpok* genes and focuses analysis on 18 genomes, of which only 10 show *FuSpok* genes on accessory chromosomes (ACs). Given this limited sample, how robust is the claim regarding the association of *FuSpok* genes with ACs?

4. Line 98 of the Results and Discussion section implies that the analysis was conducted to explore functional divergence of *Spok* genes within lineages. However, the following paragraph focuses more on differences in distribution across species rather than on functional divergence. Consider rephrasing the opening line for accuracy.
5. The statement that "The effect of decreased temperature on *Spok2* and *Spok3* suggests that this condition stabilizes the antidote function for long enough that the spores survive before both toxin and antidote function eventually degrade" is speculative. Please provide a reference or clarify that this is just an assumption.

July 2025

Dear Editor,

We thank the reviewers for their time spent reviewing our manuscript. We have addressed the concerns raised. In particular, we have reworked the statistical analysis for Figure 1 and provided more context as to the limited number of genomes we could use for this analysis. Note that this also involved removing some duplicate genomes that had been previously missed, which results in fewer tips in figure 2. None of these changes impacted the overall conclusions of the manuscript. In addition, we spent some time reworking our phrasing in our verbal model for the coevolution of *Spok* genes, ACs, and *Starships*. We have also excluded the section on the temperature of spore killing in *Podospora* and made changes to the figures to improve clarity. You will find our specific answers to the reviewer comments below.

It is our hope that the article is now ready for publication in *Genetics*, but please do let us know if additional changes are required.

Aaron Vogan

Our responses are given below in bold

Editor: Two experts in the field have reviewed your manuscript, and I have read it as well. I am pleased to inform you that, with minor revisions, it is potentially suitable for publication in GENETICS. The reviewers have comments and concerns that need to be addressed in a revised manuscript. You can read their reviews at the end of this email.

It is most important that you address the following in your resubmission:

The Reviewers have indicated several areas to strengthen the manuscript and provide some context that might temper the conclusions, specifically that they raise question that while there is statistical support for the FuSpok gene and AC relationships, there are only a limited number of genomes showing this pattern. There is concern that there may not be enough examples so that either some comments on the limitations of the current sample size or provide other context to the strength of these interpretations in the conclusions.

There was also a concern raised about the rationale for inclusion of the specific hypothetical genes and the multiple growth media that needs some more justification or context.

The integration/comparison to the Podospora results I imagine are connected based on previous work in this field and from your labs but the transition to this inclusion or context needs more description or linkage.

Reviewer #1

This paper presents a comprehensive overview and phylogenetic analysis of Spoks, a class of toxin/antidote genes in fungi, in the fungal genus *Fusarium*. Enrichment in accessory regions is shown as well as presence in 'Starships' (giant transposons) for a subset of Spok genes. A screen in yeast demonstrated toxicity of two of the six tested Spok genes (in which the antidote part was mutated). A model is presented for how Spok genes can stabilize accessory chromosomes by killing cellular compartments without them.

This is a well written paper about a very exciting topic. I have the following suggestions for improvement.

1. Concerning the concept of causality. Mention is made of a possible 'causal connection' between accessory chromosomes/regions and FuSpoks (p. 4 line 50) and, after showing enrichment of Spoks in accessory regions, it is stated that "it does not establish causality". I would like to see some clarification of what could be causing what and what the nature of this causality is. I am presuming what is meant is causality in terms of selection, not a mechanistic causality. So Spoks would not 'cause' accessory chromosomes/regions to emerge but can act to maintain them in evolution. In relation to this, on page 9 line 1-2 it is stated: "Selection may then lead to the movement and/or de novo creation of genes with roles in ..." which is a bit confusing because selection does not 'lead to movement and/or de novo creation' but only acts to maintain 'moved' or 'de novo created' genetic elements.

We have addressed this comment in two ways. Firstly, we have changed “causal connection” for “inherent connection” (Page 4 Line 31) in the section mentioned. We have also made some changes to the section on Model for the maintenance of accessory chromosomes by *FuSpok* genes in *Fusarium* and On the origin of accessory chromosomes in *Fusarium* to clarify how we imagine the vegetative action of *Spoks* could aid the establishment and maintenance of ACs. Among other things, we have rephrased the sentence of selection leading to movement or de novo creation of genes, which the reviewer pointed out.

2. Concerning the concept of orthologs. It is stated that "Many of these *FuSpoks* were very similar in sequence similarity [*see below] and likely represent orthologs (p. 5 line 4). I am not sure if the definition of ortholog is correctly applied here. In my understanding, 'orthology' denotes a relationship of descent between genes in different species, from the same ancestral gene in the last common ancestor, implied by synteny. Sequence similarity is not a primary measure of orthology, and gene expansions make orthology quite hard to assess.

Orthology can also be applied to refer to the same gene *within* a species/population, which is the intended usage here. We have modified the sentence to clarify this (Page 4 Line 116).

3. Accessory versus contig edge (Fig. 1). I understand that an arbitrary definition of 'contig edge' of 500 kb was used but now contigs 7 and 9 seem to be only 'contig edge' and not 'accessory' which gives a false impression. I do not see a solution but perhaps it suffices to state this in the text and/or figure legend.

We have changed the color coding so that the ACs are still colored as accessory in Figure 1, unless they are below 500 kB. We have also added an explanation in the legend. Furthermore, to avoid the confusion, we have separated our rings in Figure 3 to illustrate contig edges and small contigs separately from Core vs AC identity.

4. *FuSpoks* may have been introduced to *F. poae* by a starship from *F. oxysporum* Starship (p. 5 line 27-28). Similarity between the respective starships would make this plausible. Is that the case?

We have manually checked the relevant *Starships* and there is no larger pattern of synteny. However, as we point out, (Page 5 Line 16) the *Spok* genes appear to move among *Starships* relatively rapidly. Supplementary Table S2 now contains a column with the *Starship* family to emphasize this.

5. The legend in Fig. 2 mentions a 'highlight' for the clade containing the majority of *F. poae* *FuSpok* genes: I did not notice this highlight in the figure.

This highlight was included in a previous version of Fig. 2, but removed prior to submission for aesthetic reasons. We mistakenly did not update the Fig. 2 caption, and we thank the reviewer for spotting the inconsistency. We feel that the colour annotation on its own makes it clear where the large clade of *F. poae FuSpoks* are.

6. Concerning Figure 3: It would be better in my view if the title of this figure states the conclusion of the experiment. A legend should be added explaining the experiment.

We have edited the legend for Figure 3 accordingly: “Mutated copies of three *Spok* genes kill in *S. cerevisiae*. Photos show spots of yeast growth after two (Glucose) or three days (Galactose) at 30 degrees C. Left panel (Glucose) is the control and shows yeast carrying, but not expressing, each gene noted on the left of the photo. The right panel (Galactose) shows yeast growth when expressing each gene.”

7. The section on temperature and spore killing seems unnecessary. It could just be mentioned in the Discussion if at all necessary and presented in a separate paper.

We have now removed the section of temperature effects on spore killing in *Podospora*.

8. Textual issues:

- Amounts > numbers? (p. 1 line 3) **Replaced**
- Benign media (p. 1 line 41) > is this a correct term? **Changed to “permissive”**
- Phylogeny 'separated' > 'divided' (p. 4 lines 105 and 107) **Changed**
- Remove comma's in p. 4 line 117 and p. 9 line 38 **Done**
- P. 5 line 3: delete 'similarity' **Done**
- P. 7 line 68: Temperature effects > affects? **This section is removed**
- P. 9 line 21: directly > strictly? **Changed**

Reviewer #2

Comments for the Author:

1. What is the functional and biological relevance of the hypothetical genes NECHADRAFT_82228, FOXG_14281, FOXG_14774, FOXG_07107 used in the study?

We refer to section *FuSpok* genes function as toxin/antidote proteins (Page 7 Line 23) where we state: “We chose genes based on a combination of factors, including phylogenetic diversity and genomic location”. To this, we have added that *Fusarium oxysporum* f. sp. *lycopersici* is of interest because of it being well studied, and that one of our chosen *FuSpoks* reside on the chromosome previously shown to horizontally transfer between strains, and another one is found within a *Starship*.

2. In the growth assay, both glucose and galactose media were used. However, the rationale for using two types of media is unclear. Since the glucose data do not appear to add any relevant insight, consider whether those results could be omitted or more clearly justified.

To aid interpretation, under the Methods section, subsection Growth assay, we added “...with either 2% glucose (**control, as genes are not expressed on this media**), or galactose (**on which the pGal promotor is induced and the genes are expressed**).” (Page 4 Line 14) In addition, we have expanded the legend for Figure 3 so that the results should be interpretable without knowledge of the methods.

3. Out of the 146 genomes analysed, what proportion lacked any FuSpok genes and how does it impact the overall interpretation of the hypothesis?

The study states that not all *Fusarium* strains contain FuSpok genes and focuses analysis on 18 genomes, of which only 10 show FuSpok genes on accessory chromosomes (ACs). Given this limited sample, how robust is the claim regarding the association of FuSpok genes with ACs?

49 of the 149 genomes investigated did not carry *FuSpoks* that passed our filtering criteria. Based on this and other reviewers' comments, we discussed alternative ways to compare the presence of ACs and *FuSpoks*. The biggest limitation for this analysis is the lack of annotated ACs for many of the genomes investigated, which we have now highlighted (Page 4 Line 66). Additionally, we revisited our genomes and were able to identify annotated ACs in six more genomes, bringing our sample size up to 24. For these 24, we have also added a regression analysis showing a positive association between the size of the accessory genome (as a percentage of genes residing on it) and the number of *FuSpoks* each genome carries (Figure 1B).

4. Line 98 of the Results and Discussion section implies that the analysis was conducted to explore functional divergence of Spok genes within lineages. However, the following paragraph focuses more on differences in distribution across species rather than on functional divergence. Consider rephrasing the opening line for accuracy.

We have removed the two sentences on functional divergence of *Spoks* in *Podospora* and instead start the paragraph with "To investigate patterns of gene family diversification of the *FuSpoks*..." to improve clarity (Page 4 Line 91).

5. The statement that "The effect of decreased temperature on Spok2 and Spok3 suggests that this condition stabilizes the antidote function for long enough that the spores survive before both toxin and antidote function eventually degrade" is speculative. Please provide a reference or clarify that this is just an assumption.

We have removed the section on the effect of temperature on spore killing in *Podospora*.

Reviewer #3

The role of toxin/antidote genes in the maintenance and evolution of accessory chromosomes in *Fusarium*

The authors bring together two fascinating areas of genome biology, (1) gene drive, mediated by spore killing genes, Spok genes, and (2) accessory chromosomes. Fungi are an important group with which to study these processes because of the range of genome conformations they maintain, combined with their ability to invade new environments. All of this work is set against a backdrop of Starship TEs.

This is an exciting frontier of biology where little is known and we think the authors bring together some nice evidence for their model on the maintenance of Spok genes, usually implicated in spore killing, providing here a novel mechanism for their maintenance during vegetative growth. Being such a novel area of biology, this model is an important step providing something for other researchers to test, confirm/reject. However, some uncertainty in our ability to decipher figure 2, and the use of language linking the maintenance of Spok genes specifically with accessory chromosomes, which perhaps overstates causality, needs further consideration. This is in part due to the number of analyses/datasets included. The authors also introduce work from *P. anserina* as well as *Fusarium*, with temperature experiments and phenotypes in *Saccharomyces cerevisiae* - this could have been several papers, split in a number of different ways. Our comments focus on a clarification of the relationship between genome quality (ability to assemble ACs and the observations based on those ACs), clarifications around figure 2, as well as over assertion of the intrinsic role of Spoks for maintenance of accessory chromosomes, as they are absent in some of them.

Genome quality

To what extent is the observation of Spok gene presence/density in core and accessory chromosomes impacted by the quality of the genomes in the set? There was a reasonable amount of filtering from 146 genomes down to 94, and in figure 1 just 18 species were used. Across these 94 genomes there was presumably varying degrees of completeness. One would expect that completeness has a disproportionate effect on the quality of ACs. Does the Spok presence / density between core and accessory content relate to genome quality? For example, the authors find that 3/4 of Spoks are in ACs but to what extent do the authors have complete AC assemblies across their tree? What are the relative densities of Spok genes only on the core genomes, and is there a relationship between those densities, and the genomes for which they have well assembled ACs?

Indeed, the filtering down of the genomes from the original 146 to those used in Figure 1 was due to assemblies not being done to chromosome level, or accessory regions not being annotated. We re-visited our annotations of accessory chromosomes in our collection of *Fusarium* genomes and increased the number of genomes for which there was clear information in the literature about presence (or absence) of accessory chromosomes to 24 genomes. In short, the absence of annotated accessory chromosomes does not mean that the genome does not contain accessory chromosomes, only that they were not annotated. This means we are unable to answer the questions regarding genomes without accessory regions. Apart from using long-read only and a threshold for completeness, we did not further investigate genome quality.

We have, however, re-worked the statistical analyses in Figure 1 to provide a clearer picture of the association and possible enrichment of *FuSpok* genes on accessory chromosomes in *Fusarium* genomes. First, we provide a straight-forward visualization of the relationship between the number of *Spok* genes, and the percentage of genes found on accessory chromosomes, for the genomes for which accessory chromosome information was available. This demonstrates both a clear positive relationship between the gene content on accessory chromosomes and the number of *Spoks*, as well as genomes that are outliers for this general trend. In place of the binomial tests, we have provided Fisher's Exact tests for

each genome, testing for the enrichment of *Spok* genes in the accessory parts of the genome (compared to the genome wide distribution of genes on and outside of accessory regions). We also performed the analysis only for *Spok* genes not on small contigs/contig edges/within Starships. Out of 24 genomes, 16 (and for the second analysis 14) show enrichment of *Spok* genes on accessory chromosomes.

Figure 2

Figure 2 contains a great deal of information, and we spent some time referring to it to confirm statements in the MS. Being such a key figure, it is important that it is accurate and a good deal of our uncertainty about the work could be cleared up by addressing suggestions on this figure. For instance, the statement 'the single *F. poae* FuSpok in clade III may have been transferred from *Fusarium langsethiae*' (p5 lines 38-40)' is currently hard for the reader to assess.

Firstly, we suggest reconsidering using colour for three separate categories, especially with high similarity of colours within and across legends.

We have improved the visibility of the annotation rings in the main tree figure and separated the rings used for annotation of core/accessory chromosomes as well as the annotations for contig edges/small contigs. This also better reflects that number of genomes for which accessory chromosome annotations were unavailable. We have reorganized the colours used for species designations to be easier to distinguish *Fusarium* species that are more/less closely related, based on the phylogeny added in Fig 2B of the revised version of the manuscript.

Secondly, the incongruence of the tree compared with species relationships is not clear for readers who are not intimately familiar with *Fusarium* taxonomy. It would be informative, for instance, for there to be some way for the reader to cross-check *Fusarium* species relationships with the relationships in the Spok tree. This could be represented as a species tree shown alongside the *Fusarium* legend. A visualisation of species relationships would also strengthen certain parts of the text e.g. where the difference in AC and Spok content in *F. poae* and *F. graminearum* is discussed on p4 lines 110-141; the authors may also want to cite Armer et al. 2024 (<https://doi.org/10.1016/j.funbio.2024.07.004>) to illustrate this latter point.

We have added a *Fusarium* species tree to Figure 2, based on the Armer et al. (2024) publication, to provide context for the expected relationship between the genomes used in this study.

Finally, there are also some things that need to be clarified or corrected in the figure 2 legend. We assume that 'FSSC' refers to the *F. solani* species complex (syn. *Neocosmospora*), however FSSC species are also listed separately (*F. solani*, *F. falciforme*, *F. vanettenii*, *F. keratoplasticum*, *F. solani-melongenae*), so it is not clear what this category refers to. This is also the case in supplementary figure 1. *F. oxysporum* also appears twice in the legend, and there is an 'unknown' category in the genomic location legend which we cannot see being used in the figure itself.

The duplicate *F. oxysporum* entry in the legend has been removed and clarified the identity of the *F. sp* strains.

P5ln22 main text states that ...'Spok genes from different Starship families present in different Fusarium species'. How can we see in figure 2 that these are from different Starship families? Or where can we see this information?

We have added the information about *Starship* families in Supplementary Table S2, and added a reference to it at the sentence mentioned by the reviewer.

Assertive language

The barrier of proof to causation is high and, while we agree that Spoks have a role in the maintenance and evolution of accessory chromosomes in Fusarium (in the title) the language used throughout overstates this role. We list cases for consideration below.

The authors have provided evidence that Spoks are enriched on ACs in some cases, however we question whether 10 out of 18 genomes (p4 line 91/figure 1A) is a convincing number to conclude that Spok genes are a predominant cause of AC persistence. Evidently ACs are maintained for other reasons in closely related genomes - incidentally it would be nice to order the y axis by taxonomic relationships in figure 1A. That this is not a universal mechanism is also suggested by the fact that 2 of the 6 Spoks which were evaluated in yeast were located on ACs in *F. oxysporum* but were not found to have toxicity.

To clarify, we do not claim that *FuSpoks* are the “predominant” cause of AC persistence. We present a model regarding how these elements could play a role in the formation of ACs and their initial persistence in the absence of host-beneficial genes, which is currently held as the standard model. To moderate the language, in addition to the changes listed below, we have also replaced “understanding that the *Spok* genes play an important role in AC evolution” with “understanding that the *Spok* genes have the potential to play an important role” (Page 9 Line 22).

Regarding the statistical results, we have modified the section about *FuSpok* enrichment in accessory regions. We have found annotations of accessory regions for more genomes. We now present two separate analyses for the association of *FuSpoks* and accessory regions in Figure 1: we first demonstrate a positive relationship between the size of the accessory genome and the number of *FuSpoks* each genome carries. Then, we present results of Fisher’s exact test for each of the genomes, demonstrating a significant enrichment of *FuSpok* in accessory regions in 16 out of 24 genomes.

Lastly, responding to the comment that two of the six *FuSpoks* tested in yeast did not show toxicity, we refer to the last paragraph in the section *FuSpok* genes function as toxin/antidote proteins (Page 7 Line 63) where we discuss possible reasons for the lack of toxicity. We have also added a final paragraph to further explain the dynamic nature of this gene family in the section Model for the maintenance of accessory chromosomes by *FuSpok* genes in *Fusarium* (Page 8 Line 47).

P1ln10: 'Finally, as Spok genes are mobilized by the newly described TE superfamily Starships, it suggests these TEs play crucial roles in forming accessory chromosomes and regions'. This is quite a jump from Spoks in TEs to TEs in the formation of accessory chromosomes. This paper is not about the formation of accessory chromosomes.

We have modified our language and specified “we are presenting a model for how these TEs could play a role” in the formation of ACs.

P8ln88 'but the fact that at least some Spok genes studied here are on Starships supports such a role'. Only very weakly. To do this analysis properly one would have to separate the signal from Spoks both inside and outside Starships, from that signal of the hundreds of other genes both inside and outside Starships.

We have moderated our language to clarify the speculative nature of the discussion.

The section 'On the origin of accessory chromomes in Fusarium' is a little fanciful and we would suggest a rewrite that considers more of the forces that create and maintain ACs. Perhaps even venturing into sex chromosomes, repeats and their maintenance. The authors state that a chromosomal fission could produce an accessory chromosome. Presumably the other important genes that already happen to be on that region, would also play a role in its preservation? This whole section is written without much if any attention to anything other than the impact of Spoks on genome structure. For example, p8ln81: 'Alternatively, since Spok genes are frequently found on Starship transposons (Gluck-Thaler et al. 2022), a Spok gene could be brought to a novel chromosome soon after formation via transposition of a Starship, thus precluding the need for Spok genes in the background or core genome'. This argument should be backed up by evidence that Spok incorporation via HGT is more likely to occur in an AC than a core chromosome. Where, if incorporated into a core chromosome, the Spok would have a guaranteed its chance of being transmitted vertically, without the additional requirement to now have to also ensure the transmission of an AC.

We are grateful to the reviewer for the opportunity to clarify the gaps in understanding of the origins of ACs that our model attempts to fill. To clarify, we have added a sentence at the beginning of the same paragraph: “We propose a model for how the presence of *Spok* genes could help solve this riddle. Assume that...” (Page 8 Line 79). We also clarify that “In contrast to previous models on the origin and maintenance of ACs, such as them carrying genes beneficial to the host, our model does not require specific environmental conditions or growth conditions to be met for selection to start favoring these *Spok* carrying chromosomes.” *Page 9 Line 5).

P9ln1 After Spoks are present in and maintaining the AC 'selection may then lead to the movement and/or de novo creation of genes with roles in rapidly evolving traits, like virulence, on said chromosome'. Why wouldn't any of these non-selfish processes already have occurred on the AC, and be maintaining them in their own right? This highlights the authors drive to put Spoks at the centre of every argument on the generation and maintenance of ACs. It reduces trust in objectivity.

Our aim with the paper is to present a possible explanation for the widespread distribution of these elements. We have tried to present a possible model with moderating language.

P9ln17: 'It remains to be answered whether there is a deep evolutionary link between the Starships and Spok genes or if Starships recurrently abduct Spoks'. A Starship cannot abduct anything.

This was meant as a metaphor, as the mechanism of gene acquisition by *Starships* is unknown, they may indeed “abduct sequences”. To clarify, we have rephrased the line: “It remains to be answered whether *Starships* and *Spok* genes have co-evolved since their

respective origins, or if *Spoks* are recurrently found in *Starships* by chance.” (Page 9 Line 26).

Minor comments

P. anserina work comes quite out of the blue, should this have another header?

We have removed the section on the effect of temperature on spore killing in *Podospora*, since all reviewers were confused about its inclusion.

Spore killing reduced at 22 and returned at 27. Can the authors add something in the introduction to account for the life history and environment these fungi are exposed to. These temperatures seemed very high and had us questioning how regularly these fungi are exposed to these high temperatures, at which *Spoks* have any spore killing ability.

We have removed the section on the effect of temperature on spore killing in *Podospora*, since all reviewers were confused about its inclusion.

Supplementary tables are all missing - the captions are there but no tables

We have uploaded our supplementary tables, and do not know why these were not shared to the reviewers.

P4ln52-76 - this bit feels more like pre-processing/methods than results?

While we see the reviewer’s point of view, in that we describe what we did, we believe that the issues we discovered while assembling our list of *Spok* homologs were results worthy to report by themselves: the mis-annotation of introns and the repeated erroneous calls by Nanopore.

September 10, 2025
RE: GENETICS-2025-308521

Dr. Aaron A. Vogan
Uppsala Universitet
Institute of Organismal Biology
18D Norbyvägen
Uppsala 75644
Sweden

Dear Dr. Vogan:

Congratulations, your manuscript titled "The role of toxin/antidote genes in the maintenance and evolution of accessory chromosomes in *Fusarium*" is accepted for publication in GENETICS! Many thanks for submitting your research to the journal.

The revision has addressed the reviewers points with some improvements to the clarity of the message and some changes to the figures as noted in the rebuttal letter and resubmitted manuscript.

Only one point I wanted to raise is I note in looking at the Supplemental Table 1, there are a few *Podospora* assemblies which do not have genome accession numbers, PaWa87p is linked to <https://www.ncbi.nlm.nih.gov/biosample/SAMN10977284> but this sample seems to be linked to a Bioproject <https://www.ncbi.nlm.nih.gov/bioproject/PRJNA523441> with a accession for the assembly GCA_017654855.1 as which is *Panserina* strain Wa137- accessions not Wa87+ ? and PcTp does not have an accession number for a genome.

These are minor points but can you check your supplemental table is complete and up to date?

To Proceed to Publication:

1. Format your article according to GENETICS style: <https://academic.oup.com/genetics/pages/author-guidelines>
2. Ensure that you comply with data and community resource citation guidelines: <https://academic.oup.com/genetics/pages/author-guidelines#section-5-9-2>
3. Upload your final files at <https://genetics.msubmit.net>
4. Add oupsupport@scipris.com and genetics.oup@novatechset.com (or the domains @scipris.com and @novatechset.com) to your email program's "safe senders" list. You will be contacted by both at various points during the production process.

Notes:

- Your currently-accepted manuscript (unedited, as submitted, reviewed, and accepted) will be published at GENETICS and deposited into PubMed as an Advance Access article. Notify sourcefiles@thegsajournals.org before signing your license if you do not wish to publish your article via Advance Access.
- We invite you to submit an original color figure related to your paper for consideration as cover art. Please email your submission to the editorial office or upload it with your final files. You can submit a small-sized image for evaluation, and if selected, the final image must be a TIFF file 2513px wide by 3263px high (8.375 by 10.875 inches; resolution of 600ppi). Please avoid graphs and small type.
- After files are sent to Oxford University Press we use SciPris to manage article licensing and payment. If you do not have a SciPris account, you will receive an email from no-reply@scipris.com to sign up to use Oxford University Press' author portal. After logging in, follow the online instructions to sign your license and arrange any payment due.

If you have any questions or encounter any problems while uploading your accepted manuscript files, please email the editorial office at sourcefiles@thegsajournals.org.

Sincerely,

Jason Stajich
Associate Editor
GENETICS

Approved by:
David Begun
Senior Editor
GENETICS

Review comments (if applicable):